# Cutaneous Melanoma versus Vulvovaginal Melanoma—Risk Factors, Pathogenesis and Comparison of Immunotherapy Efficacy

**DOI:** 10.3390/cancers14205123

**Published:** 2022-10-19

**Authors:** Anna Lorenz, Mateusz Kozłowski, Sebastian Lenkiewicz, Sebastian Kwiatkowski, Aneta Cymbaluk-Płoska

**Affiliations:** 1Department of Gynecological Surgery and Gynecological Oncology of Adults and Adolescents, Pomeranian Medical University in Szczecin, al. Powstańców Wielkopolskich 72, 70-111 Szczecin, Poland; 2Department of Obstetrics and Gynecology, Pomeranian Medical University in Szczecin, al. Powstańców Wielkopolskich 72, 70-111 Szczecin, Poland

**Keywords:** melanoma, immunotherapy, nivolumab, pembrolizumab, ipilimumab

## Abstract

**Simple Summary:**

Melanoma of the vulva and vagina is a relatively rare neoplasm, unlike melanoma of the skin. Its prognosis is poor, and its pathogenesis is not fully understood. Immunotherapy is one of the rapidly developing cancer treatment methods. In this article, we focus on the pathogenesis of lower genital tract melanomas and related risk factors and compare the effectiveness of two groups of drugs—anti-PD-L1 and anti-CTLA4 antibodies—in the treatment of this condition. This type of immunotherapy is a relatively common treatment method for cutaneous melanoma but not for the rare vulvovaginal melanoma. For vulvovaginal melanoma, the effects of these treatments appear to be limited; however, this requires further research.

**Abstract:**

Cutaneous melanoma is a relatively common neoplasm, with fairly well understood pathogenesis, risk factors, prognosis and therapeutic protocols. The incidence of this disease is increasing every year. The situation is different for rare malignancies such as vulvar melanomas and for the even rarer vaginal melanomas. The risk factors for vulvovaginal tumors are not fully understood. The basis of treatment in both cases is surgical resection; however, other types of treatments such as immunotherapy are available. This paper focuses on comparing the pathogenesis and risk factors associated with these neoplasms as well as the efficacy of two groups of drugs—anti-PD-L1 and anti-CTLA4 inhibitors—against both cutaneous melanoma and melanoma of the lower genital tract (vulva and vagina). In the case of cutaneous melanoma, the situation looks more optimistic than for vulvovaginal melanoma, which has a much worse prognosis and, as it turns out, shows a poorer response to immune therapy.

## 1. Introduction

Melanoma is a malignant skin neoplasm that derives from melanocytes. Melanocytes are a type of pigment cells located in the basal layer of the epidermis, responsible for the production of the pigment melanin. These specific cells, however, are not exclusively located in this tissue. Melanocytes can also be found in the mucous membranes of the genito-urinary tract, airway tract and even in the gastrointestinal track. Melanoma derived from this type of tissue is called mucosal melanoma. Every year, there are between 160,000 up to 230,000 new cases of melanoma worldwide. The mortality rate worldwide is between 48,000 and 55,000 yearly [1,2]. Since the 1970s, the number of new cases has increased both in women and in men [1]. The 5-year relative survival is 74% in eastern Europe, rising to 82% in southern Europe and up to 88% in Scandinavia and central Europe. For the United Kingdom and Ireland, the relative survival rate is 86% [3]. These data indicate a relatively good prognosis. The risk factors for melanoma include both extended and brief exposures to UV radiation (most probably including artificial UV sources) [4], a fair skin complexion and bright eye color, freckles developing before the age of 15, red or blonde hair [5], numerous skin lesions including atypical lesions [6], melanoma cases in family history [7], melanoma in patient’s history [8], a low socio-economic status [9]. Cutaneous melanoma can be divided into four distinctive main types: superficial—most commonly appearing on skin exposed to short-term, but intensive UV radiation—nodular melanoma—deriving from de novo lesions mostly on the torso, head and neck—lentigo maligna—commonly on skin exposed to long-term UV radiation—acral lentiginous—on hairless skin of feet, hands and underneath the nails. This neoplasm is known to easily create metastases, including satellite-type metastases, in-transit metastases, lymph nodes metastases and metastases spreading via the blood and becoming established mostly in the lungs, brain, liver and bone. 

On the other hand, mucosal melanoma presents different features. It is a relatively rare form of neoplasm—only 1.5% of all melanoma cases, which amounts to around 0.03% of total neoplasm cases. It is mostly located in membranes of the head and neck area (55.4% of all mucosal melanomas), genital tract (18%), rectal area (23.8%) and urinary tract (2.8%) [10]. Its prognosis is unfavorable. The 5-year survival rate of patients with mucosal melanoma is 25% but drops to 11.4% in case of melanoma of the genital tract [10]. 

As it turns out, melanoma of the skin are detected faster than melanomas of the vulva and vagina (Table 1 and Table 2) [11,12,13].

For cutaneous and genital melanomas, surgical resection remains a basic form of therapy. Depending on the stage of the disease, other forms of therapy such as radiotherapy, chemotherapy and immunotherapy are available. The main aim of this study was to compare the effectiveness of immunotherapy for cutaneous and genital tract melanoma.

## 2. Pathogenesis

About one half of skin melanomas arise from de novo lesions, and the other half from malignant pigmented nevi [14]. Various factors, both exo- and endogenous, influence the process of formation of tumor cells as well as their further expansion. The involvement of the immune system is also important. The proliferation rate of mutant melanocytes increases significantly [15]. The most prominent external factor that promotes such process is UVA and UVB radiation. Internal factors play a role in cutaneous melanoma pathogenesis, including mutations such as *BRAF, NRAS, TP53, CDKN2A* and *PTEN* [16,17]. A *BRAF* gene mutation (variant V600E) is found in roughly 50% of primary skin melanomas; however, the role of this mutation in metastatic melanoma is yet to be determined [18]. The *BRAF^600^* mutation coexisting with mutations in the *CDKN2A* gene impairs the control over the entire cell-cycle. Additional mutations, e.g., in *PTEN* and *TP53* favor a more aggressive growth and expansion of this neoplasm and promote the formation of metastases, including distant ones [14,16]. Another important mutation involves *KIT.* It is also common in acral melanoma [19]. The common changes regarding *KIT* are amplifications and missense mutations in the auto-inhibitory domain (coded by exon 11) and the tyrosine kinase domains (coded by exons 12–21). Those mutations are the most important non-synonymous *KIT* mutations [20].

The spliceosomal protein *SF3B1* is a core component of the U2 snRNP. This protein’s main function is in the splicing process, during which non-coding pre-mRNA fragments are removed thus producing a target product, i.e., a proper mRNA. U2 snRNP targets the branch point sequence at the 3′ splice site of an exon–intron junction.

Mutations in *SF31B* result in alterations in the splicing process [21]. There are two possible outcomes for such scenario. The first outcome leads to a translation of a transcript, thus producing an aberrant protein. The second scenario leads to a nonsense-mediated decay (NMD), which results in mRNA and protein downregulation. 

Across neoplasm types, *SF3B1* mutations have been identified as heterozygous and regard, in particular, R625, K666 and K700 residues. Mutations that are characteristic for mucosal and uveal melanomas almost exclusively occur at R625. 

Hintzsche et al. [22] documented the presence of *SF3B1* mutations in 35% (7/19) of mucosal melanoma cases. They were most commonly observed in anorectal melanomas (3/5, amounting to 60%) and in the vulvovaginal melanomas (4/9, amounting to 44.4%).

Quek et al. [23] provided similar results by showing that *SF3B1* mutations mostly occurred at *SF3B1-R625* (5/6, amounting to 83%) and were almost exclusively found in vulvovaginal (5/19, amounting to 26%) and the anorectal melanomas (3/5, amounting to 60%). The following graphic (Figure 1) shows collectively the most common mutations leading to melanoma development.

Mutations in specific proteins disrupt the function of cellular pathways, such as the MAP kinase, B cell kinase (AKT), PI3K–mTOR kinase, or the PTEN pathway [24,25,26]. In addition, impaired immune response mechanisms are involved in the pathogenesis of cutaneous melanoma. Under the influence of interferons (including γ, as well as α or β), there is excessive activation of the JAK1/2 and STAT kinase pathways, which stimulate the production of the PD-L1 and PD-L2 ligands. These ligands are transported to the cell membrane and are presented to T lymphocytes. On the surface of these lymphocytes are PD-1 (programmed-cell-death protein-1) checkpoints, which protect the cells from an excessive immune system response which would destroy them. In this way, cutaneous melanoma cells escape the control of T cells [27,28]. Interferons may have different functions, such as anti-angiogenic, anti-proliferative as well as immunomodulatory effects, which may be crucial in the development of melanoma [29]. The following graphic illustrates the effect of anti-PD-L1 antibodies that prevent PD-1 from binding to its ligand (Figure 2).

In the case of genital tract melanoma, in particular of the vulva and vagina, the situation is different. In fact, the risk factors are not fully understood, and, in comparison to skin melanoma, this neoplasm occurs much less frequently—it accounts for 3 to 7% of all melanomas [30] and only for 10% of all vulvar malignant neoplasms [31]. Among the risk factors is advanced age (average age for this melanoma is over 60 years) and white race [32,33,34]. The vulva area is not exposed to UV radiation, so this exogenous risk factor can be ruled out [33].

Vulvar melanoma is most commonly located on the labia majora (around 51%) and labia minora (around 38%); the least common location is the clitoris (around 12%) [33].

The genetic profiles of cutaneous melanoma and vulvar and vaginal melanoma are different in terms of mutations. The *KIT* gene mutation, which rarely appears in cutaneous melanoma patients [35,36], appears to be commonly occurring in vulvar melanoma (from 22 up to 31% of cases). It is more common in vulvar melanoma (27%) and less common in vaginal melanoma (8%) [37,38]. On the other hand, *NRAS* mutation occurs in around 5.3% to 12% of genital tract melanomas [38,39], with a higher incidence in vaginal melanomas [40]. 

There are studies suggesting that the occurrence of the *NRAS* mutation might even be more common than that of the *KIT* mutation [41]. The *BRAF* mutation seems to be less frequent in vulvar melanoma [37,38] than in cutaneous melanoma (where it is present in 60–63% of cases)) [16,18]. As regards protection against T cell activation, the defense mechanism in genital tract melanoma is similar to that in cutaneous melanoma: the PD-L1 ligand is produced and binds to PD-1 T cell checkpoints on the surface of the tumor cells; this mechanism is also relatively common in vulvar melanoma [29,42,43]. In addition to the previously described mechanism, cells have also a CTLA-4 checkpoint which, upon associating with specific ligands (e.g., PD-L1) blocks the immune reaction mediated by T cell lymphocytes [44].

## 3. Immunotherapy of Cutaneous Melanoma

The primary treatment for both cutaneous and vulvar melanoma is surgical resection. Other alternative treatments include radiation therapy, chemotherapy and immunotherapy. Different drugs are used in immunotherapy for cutaneous melanoma, but the focus still remains on two groups, i.e., anti-PD-1 antibodies (nivolumab and pembrolizumab) and anti-CTLA-4 antibodies (ipilimumab) (Table 3). According to different studies from different medical centers, the effectiveness of treatment varies. Larkin J. et al. [45] conducted an analysis of the treatment efficacy of nivolumab and ipilimumab in monotherapy and in combination therapy in patients with stage III and IV disease. With the combination therapy, the median disease progression-free period averaged 11.5 months compared to 6.9 for nivolumab monotherapy and 2.9 for ipilimumab monotherapy. In the presence of the PD-L1 ligands from the combination therapy and the nivolumab monotherapy, the median progression-free period was 14 months, while in the absence of these ligands, it was 11.2 months (combination therapy) and 5.3 months (nivolumab). The response rate for the combination therapy was 57.6%, 43.7% for the nivolumab group and 19% when ipilimumab was administered. Another study (Wolchok et al.) [46] examined the response in patients with advanced melanoma receiving different doses of a combination therapy with nivolumab and ipilimumab. The largest reduction in tumor volume after 12 weeks of treatment occurred at the doses of nivolumab of 1 mg/kg and of ipilimumab of 3 mg/kg (41%): these doses also promoted the largest partial and objective response (53%) to treatment. The lowest partial and objective (21%) response to treatment occurred with nivolumab at a dose of 0.3 mg/kg and ipilimumab at a dose of 3 mg/kg. In conclusion, the combined therapy and nivolumab are more effective than ipilimumab alone.

A team under the same leadership [47] performed another analysis of the efficacy of nivolumab and ipilimumab in patients depending on the presence or absence of *BRAF* mutations in previously untreated advanced melanoma. The patients were divided into three groups: one receiving the combination therapy (group 1), the second receiving the nivolumab monotherapy (group 2), and the third administered the ipilimumab monotherapy (group 3). The period without disease progression (assumed to be a cutoff point at 36 months) was the longest for group 1 (39%), followed by group 2 (32%) and group 3 (10%). Objective response to treatment was the highest in group 1 (58%), followed by group 2 (44%) and finally group 3 (19%). The overall survival rate was analogous: the rate was the highest in group 1 (58%) and the lowest in group 3 (34%). Based on this study sample, it can be concluded that nivolumab is more effective in the treatment of cutaneous melanoma than ipilimumab. Larkin et al. [48] analyzed the 5-year follow-up of patients with advanced melanoma (previously untreated) undergoing immunotherapy and divided in three groups: group 1 with combination therapy, group 2 with nivolumab monotherapy and group 3 with ipilimumab monotherapy. The presence or absence of *BRAF* mutations was also considered. The overall survival rate at 60 months was the highest in group 1 (60%) and the lowest in group 3 (26%). The response to treatment was the highest in group 1 (58%), followed by group 2 (45%) and finally group 3 (19%). The presence of a *BRAF* mutation was not insignificant: in group 1, the survival rate of patients with this mutation was 60%, versus 48% for patients without the mutation. A smaller difference was noted in group 3, i.e., 30 versus 25%. In conclusion, the combined therapy seems to be more effective than the monotherapy.

A comparative analysis of the efficacy and overall survival of patients with advanced-stage (metastatic) melanoma treated with either nivolumab or dacarbazine was also performed. Patients with *BRAF* mutation were excluded [49]. After one year of treatment, the overall survival rate was 72% for patients receiving nivolumab and 42% for those receiving dacarbazine. The response to treatment differed significantly, being 40% for nivolumab-treated patients and only 13.9% for dacarbazine-treated patients. The median survival also differed significantly: 5.1 for patients treated with immunotherapy and 2.2 for those receiving dacarbazine. The presence of PD-L1 ligands was nearly the same in the two groups. i.e., 35.4% of patients revealed the presence of these ligands. This study indicates a higher efficacy of nivolumab in comparison to dacarbazine.

The effectiveness of immunotherapy in melanoma patients carrying the *CDKN2A* mutation was investigated [50]. All patients had advanced-stage cancer, and some of them also carried *BRAF* mutations (V600E or V600K variant, 73%). It was found that 58% of the patients achieved a partial response to treatment, and 32% achieved a complete response. This is a very good result; however, the relatively small study group of 19 patients should be taken into account. Compared to other treatments, partial and complete response rates were higher in patients with this mutation; however, the mechanisms responsible for this are not fully understood.

The efficacy of immunotherapy against melanoma and other cancers was examined in a larger group of patients carrying various *CDKN2A* major loss-of-function (LOF) mutations. The presence of the wild-type gene was also investigated [51]. The results showed that patients with LOF mutations in bladder cancer had a lower median survival and overall survival rate compared to patients with the wild-type gene. However, there was no association between the LOF mutation and the efficacy of immunotherapy for melanoma and esophageal or lung cancers. Horn et al. investigated the presence not only of *CDKN2A* mutations, but also of abnormal JAK2 kinase function. This study showed that tumors with these two concomitant abnormalities, may be susceptible to developing resistance to treatment with interferon and immunotherapy [52]. On the other hand, a study showed that the presence of *CDKN2A* and *TP53* mutations did not affect the efficacy of immunotherapy. The response rates of patients with a *TP53* mutation or the wild-type gene were 47.4% and 34.3%, respectively, at *p* = 0.15; for patients with a *CDKN2A* mutation or the wild-type gene, the response rates were 45.7% and 36%, respectively, at *p* = 0.54 [53]. Patients with cutaneous melanoma or melanomas of unknown origin were included in this study.

An analysis of the efficacy of treatment in patients with or without *NRAS* mutation was performed [54]. The response to treatment in both cases was similar, i.e., 13 to 15% for patients treated with ipilimumab, 21% to 13% (wild type) with for patients receiving anti-PD-L1 antibodies and 40% to 39% (wild-type) for patients treated with combination therapy, but the median survival was higher for patients without an *NRAS* mutation (33 months versus 21 months). The study showed that a prior treatment of melanoma with MEK inhibitors likely has a favorable effect on patient prognosis. Hu-Lieskovian et al. [55] investigated in vivo the treatment efficacy of anti-CTLA4 antibodies with the simultaneous administration of *BRAF* inhibitors and/or MEK inhibitors in melanoma patients with the *BRAF* (V600E) mutation. The highest efficacy was measured for the triple-drug therapy: within 30 days, the tumor area did not exceed 25 mm^2^; however, this was a preclinical experiment that needs to be repeated in a clinical trial. 

The large clinical trial KEYNOTE-054 [56] compared the survival of patients with stage III cutaneous melanoma (IIIA, B or C) treated with pembrolizumab to that of patients receiving a placebo. All patients included in this study had histologically confirmed metastases in regional lymph nodes. It was required to check PD-L1 expression in tumor sample. After three years, the survival rate of patients in the pembrolizumab group was 63.7%, and that of patients in the placebo group was 44.1%. The KEYNOTE-006 study [57] compared the efficacy of pembrolizumab (different doses and dosing regimens) to that of ipilimumab. The patients included in this study had ipilimumab-naive, unresectable stage III or IV melanoma. The results showed that pembrolizumab was more effective than ipilimumab (OS = 32.7 months vs. 15.9 months) and was also associated with a better quality of life after 3 months of treatment [58]. The survival rates after 24 months of therapy with pembrolizumab and ipilimumab were also examined. The survival rate in patients previously untreated was higher in the pembrolizumab group (31%) than in the ipilimumab one (14.6%). In relation to the presence of PD-L1 ligands, similar results were obtained, i.e., 33.2% to 13.1% [59]. The KEYNOTE-002 trial compared the efficacy of pembrolizumab to that of chemotherapy in patients with ipilimumab-resistant melanomas. The patients included in this clinical trial had histologically or cytologically confirmed stage III or IV melanomas, which were unresectable. The overall survival was higher for patients treated with pembrolizumab than for those who received standard chemotherapy [60,61,62].

## 4. Immunotherapy of Vulvar and Vaginal Melanoma 

As it is a relatively rare cancer mainly treated by surgical resection, large clinical trials as those for cutaneous melanoma (KEYNOTE type) have not been conducted to date (Table 4). Hou et al. [37] conducted an analysis of different genes expression variants in vulvar melanoma and cutaneous melanoma. The vulvovaginal melanoma study group was much smaller than the cutaneous melanoma group. It turned out that the *BRAF* mutation rate was higher than that reported in the literature (in this study *BRAF* rate was 26%). The *KIT* mutation rate was 22%. What is interesting is that the *KIT* mutation rate was higher in vulvar melanoma (31.4%) than in vaginal melanoma (6.2%). The authors demonstrated that 75% of the patients showed PD-1 expression, and 56% showed PD-L1 expression, which suggests a therapeutic option in the form of immunotherapy for these patients. 

Albert et al. [63] took into account the contribution of immunotherapy to treatment. The overall survival rate was higher for patients who received the monoclonal antibodies but statistically insignificant. In this study of more than 1900 patients with melanoma in the genital tract, patients with in situ tumors were excluded. 

In a combined analysis of six clinical trials [64], patients with vulvar melanoma received nivolumab as monotherapy (86 patients) or in combination with ipilimumab (35 patients). The progression-free period was shown to be similar to that of patients with other mucosal melanomas, but the response rate was lower (37%) than that observed in patients with cutaneous melanoma (60%). An even lower treatment effect (19% response rate) was shown for patients with mucosal melanoma in the KEYNOTE 001, 002 and 006 clinical trials [65].

Egger et al. [66] studied 20 patients with melanoma of the lower genital tract (14 vulvar and 6 vaginal). Nine patients received immunotherapy (the average response time was 4 months), and three received immunotherapy in combination with radiation therapy (the response appeared after an average of 5 months). The treatment of five cases of vaginal melanoma [67] in which nivolumab was administered at recurrence (in one patient undergoing monotherapy, in the other in combination with dacarbazine) was described retrospectively. When comparing the overall survival of these patients to that of patients who refused further treatment at recurrence, the treatments administered likely prolonged patients’ survival. Wilhite et al. [68] reported that vulvar and vaginal melanomas showed a lower expression of adaptive immunity genes and a lower expression of PD-L1 ligands compared to cutaneous melanoma, which probably caused a worse prognosis after immunotherapy.

Skovsted et al. [69] described a case series of vulvar and vaginal melanomas; however, the number of patients was too small to assess the efficacy of the treatment, which, perhaps in combination with resection, could have a significant impact on patient survival. Boer et al. [70] described 198 cases of vulvar melanomas. The response rate to anti-PD-L1 antibodies was 18% and increased to 20% when this treatment was combined with anti-CTLA-4 antibodies; however, the study group was too small for far-reaching conclusions.

In a small study group of six patients, ipilimumab was used as therapy. The survival rate after one year was 33% in this group of patients [71].

Chlopik et al. [72] examined in their study the prognostic role of the expression of the PD-L1 in the tumor and of CD8+ and FoxP3+ in lymphocytes in vulvar melanoma patients. The study group included 75 patients with primary vulvar melanoma. The authors found that the peritumoral expression of CD8+ and the tumoral expression of FoxP3+ lymphocytes were associated with better overall survival (*p* = 0.021 and *p* = 0.0055 respectively). The same results were reported in regard to PD-L1 expression > 5% (*p* = 0.03).

Indini et al. [73] investigated the results of immunotherapy against tumors in the lower genital tract. In this study, 29% of the patients had metastatic disease from the start, and the rest developed metastases during the treatment. As for the treatment, 57% of the patients received ipilimumab, while 43% were administered anti-PD-L1 antibodies. The response rate reached 28.5%. The patients in the anti-PD-L1 antibodies group had a better progression-free survival.

Several cases have also been published in which immunotherapy likely prolonged patients’ lives, but the use of this type of treatment requires further research in the future [74,75,76].

## 5. Conclusions

Unlike skin melanoma, melanoma of the lower genital tract is a rare and therefore little understood cancer. A comparison of the effectiveness of common treatments against vulvovaginal melanoma and cutaneous melanoma, it appears that both survival and recurrence-free period are shorter for patients with melanoma of the lower genital tract. Not all molecular mechanisms involved in the pathogenesis of vulvar melanoma have been understood. This pathogenesis of this tumor as well as the efficacy of immunotherapy against it, require further study.

## Figures and Tables

**Figure 1 cancers-14-05123-f001:**
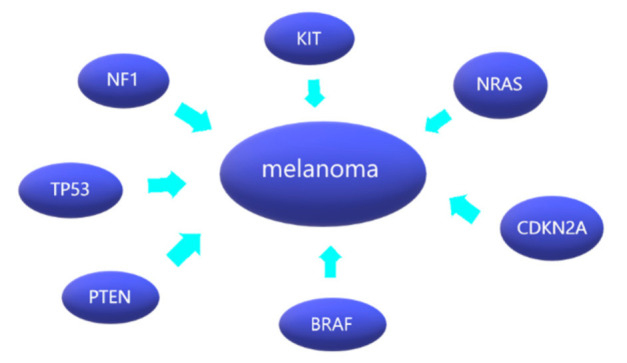
Melanoma and its most common mutations.

**Figure 2 cancers-14-05123-f002:**
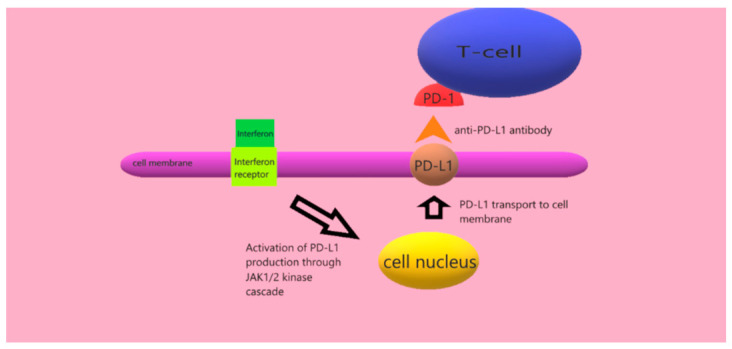
Effect of anti-PD-L1 antibody.

**Table 1 cancers-14-05123-t001:** Cutaneous melanoma; stage at diagnosis and 5-year survival.

Survival (%)	Diagnosis (%)	Stage of Neoplasm in SEER Staging
95	70	I
70	15.2	II
46	9.8	III
12	5	IV

**Table 2 cancers-14-05123-t002:** Vulvar and vaginal melanoma; survival and diagnosis rate based on staging. Table designed and based on work of Wohlmuth et al. [12].

Vaginal Melanoma (%)		Vulvar Melanoma (%)		
Overall Survival	Percent of Diagnosed with Given Stage	Overall Survival	Percent of Diagnosed with Given Stage	Stage of Neoplasm in SEER Staging
23.3	36.7	70.6	52.6	localized
21.2	21.4	35.4	24.9	regional
6.3	25.1	13.4	6.7	distant
17.9	16.8	59.4	15.8	unstaged

**Table 3 cancers-14-05123-t003:** Effects of immunotherapy in patients with cutaneous melanoma. Nd—no data; WT—wild-type; T-N—treatment-naive; P-T—previously treated.

Treatment Response	Progression-Free Survival (Months)	Median Overall Survival	References
%			
43.7 (nivolumab)	6.9 (nivolumab)	nd	Larkin et al. [37]
57.6 (combined)	11.5 (combined)		
19 (ipilimumab)	2.9 (ipilimumab)		
40	nd	nd	Wolchok et al. [38]
44 (nivolumab)	nd	37.6 (nivolumab)	Wolchok et al. [39]
58 (combined)		19.9 (ipilimumab)	
19 (ipilimumab)		not reached (combined)	
45 (nivolumab)	6.9 (nivolumab)	36.9 (nivolumab)	Larkin et al. [40]
58 (combined)	11.5 (combined)	19.9 (ipilimumab)	
19 (ipilimumab)	2.9 (ipilimumab)	not reached (combined)	
40 (nivolumab)	5.1 (nivolumab)		Robert et al. [41]
13 (dacarbazine)	2.2 (dacarbazine)		
58	nd	nd	Helgadottir et al. [42]
Statistically insignificant	nd	27.2 (with mutation)	Adib et al. [43]
		not reached(without mutation)	
47.4 (TP53 mut) vs. 34.3(TP53 WT)	nd	8.0 (TP53 mut) vs. 6.0 (TP53 WT)	DeLeon et al. [45]
45.5 (CDKN2A mut) vs. 36% (CDKN2A WT)		14.0 (CDKN2A mut) vs. 6.0 (CDKN2A WT)	
Ipilimumab—15 (mutation) vs. 13 (WT)	3	21.0 (NRAS mutation)	Kirchberger et al. [46]
monotherapy anti-PD-L1—21 (mutation) vs. 13 (WT)		33.0 (NRAS wild type)	
combined 40 (mutation) vs. 39 (WT)			
	RFS after 3-years—63.7% (pembrolizumab) vs. 44.1 (placebo)		Eggermont et al. [48]
42 (combined pembrolizumab)	8.4 (combined pembrolizumab)	32.7 (combined pembrolizumab)	Robert et al. [54]
17 (ipilimumab)	3.4 (ipilimumab)	15.9 (ipilimumab)	
39.4 (pembrolizumab)	T-N: 6.6 (pembrolizumab) vs. 2.8 (ipilimumab)	T-N—not reached (pembrolizumab) vs. 17.1 (ipilimumab)	Carlino et al. [50]
13.3 (ipilimumab)	P-T: 2.9 (pembrolizumab) vs. 2.8 (ipilimumab)	P-T—23.5 (pembrolizumab) vs. 13.6 (ipilimumab)	

**Table 4 cancers-14-05123-t004:** Effects of immunotherapy in patients with vulvar and vaginal melanoma. L—localized; R—regional; D—distant; nd—no data; VVM—vaginal/vulvar melanoma, CM—cutaneous melanoma.

Treatment Response (%)	Progression-Free Survival (Months)	Median Overall Survival (Months)	References
33	nd	55.8 (L)	Albert et al. [63]
		22.2 (R)	
		5.1 (D)	
34.1	nd	19 (VVM)	Wilhite et al. [68]
		37 (CM)	
45	11(with immunotherapy)	16	Boer et al. [70]

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
