# Peer review of "Cutaneous Melanoma versus Vulvovaginal Melanoma—Risk Factors, Pathogenesis and Comparison of Immunotherapy Efficacy"

_cancers, 2022, doi:10.3390/cancers14205123_

Round 1

Reviewer 1 Report

In this review “Cutaneous melanoma versus vulvovaginal melanoma – risk factors, pathogenesis and comparison of immunotherapy efficacy”, the authors have reviewed the pathogenesis and immunotherapy efficacy of cutaneous and vulvovaginal melanoma. The tables are good and cutaneous melanoma immunotherapy has been discussed elaborately.

However, this review is not very comprehensive (many recent mucosal melanoma immunotherapy studies not discussed) and has major grammatical errors and typos. Some paragraphs are repeated (eg. Lane 155-163 repeated again from lane 163-171, Lane 171-177 repeated again from lane 178-184), the gene nomenclature is incorrect with typos and it has not been italicized. This paper needs a lot of major correction before it can be considered for publication or review.

Author Response

Dear Reviewer,

we send our response to your comments.

Kind regards,

Anna Lorenz

Reviewer 2 Report

General comments:  This manuscript compares cutaneous and vulvovaginal (VV) melanomas in terms of standard management, pathogenesis of disease, prognosis, and efficacy of immune checkpoint inhibitors in patients with advanced disease. This a potentially interesting topic because of recent advances in genomic analyses and immunotherapy for metastatic melanoma.

1.     The authors need to distinguish standard management of local disease from that of regionally advanced disease and metastatic disease.  As written, all stages of disease are blurred, and it misleadingly suggests that radiation or immunotherapy are alternatives to surgery for treatment of localized disease.

2.     The authors need to provide information on the numbers of non-synonymous mutations typically encountered in cutaneous melanoma versus vulvovaginal (VV) melanoma. While the analysis by driver mutations and loss of suppressor function mutations is appropriate, the actual number of neoantigens seems to be a more important predictor of response to immune checkpoint inhibitors.

3.     In addition to overall survival for all patients, prognosis should also be presented by stage: local, regional, distant for both cutaneous and VV melanoma, and relative frequency.  I suspect that a much higher percentage of cutaneous melanoma patients have localized disease at diagnosis compared to VV, simply because of the ease of personal suspicion of early lesions, and that stage for stage VV has a somewhat worse prognosis.

4.     The review of clinical trials reporting efficacy of checkpoint inhibitors should be better organized and divided into multiple paragraphs rather than 2 pages of one continuous paragraph, as currently written.

Specific comments:

Abstract lines 22-24.  This is misleading as written.  Surgical resection is the treatment of choice for localized cutaneous and VV melanomas, part of multimodality therapy for regionally advanced disease, and systemic therapy is the primary treatment of distant metastatic disease and regional disease that has occurred after prior therapy.  These treatments are not considered “alternatives” to surgery and “thriving” is not the appropriate translation. Radiation is often used as part of multimodality therapy for regionally advanced disease, more so in VV than cutaneous, and otherwise for palliation of specific metastatic lesions, sometimes with the concept of enhancing immunotherapy by the release of tumor associated antigens.

Abstract: It would be appropriate to comment on the relative number of non-synonymous mutations contained in cutaneous (much higher) than in VV melanomas.

Introduction: line 43: would translate better to English as “… include both extended and brief exposure to UV radiation…

Introduction: lines 55-60.  Please present survival by stage of disease for both cutaneous and VV melanomas.  I would suggest showing this in a table that includes percentage of patients by stage, and survival by stage for each, which will simplify verbiage in the introduction.  In that way it will be obvious to a reader which melanomas are more likely to be diagnosed at an earlier stage, and how survival compares within each stage.  Not a focus of this paper, but this is also an important contrast to acral lentiginous melanomas.

Pathogenesis:  Somewhere in this discussion the authors should contrast the typical numbers of non-synonymous mutations, and therefore unique patient-specific neoantigens, that are found in cutaneous melanoma compared to VV melanoma.  In addition to what we currently know about common driver mutations and loss of function (especially suppressor function) some of these likely do contribute to pathogenesis in ways we do not yet understand.

I suggest the authors read, summarize and cite Nassar KW et al. The mutational landscape of mucosal melanoma. Semin Cancer Biol 2020;61:139-148.   Includes similarities and differences among melanoma originating in oral mucosa, rectal mucosa and vaginal mucosa. It also includes  some good examples of summary statements related to comparisons among cutaneous, acral lentiginous, and various mucosal melanomas.

Pathogenesis: lines 110-112- may want to note and reference that C-kit is also relatively common in acral lentiginous melanoma and other mucosal melanomas.

Immunotherapy:  This 2-page run-on paragraph should be subdivided into several (e.g. 7) paragraphs based on the conclusion that is to be drawn from the data that is cited in each section.  References should be cited early in the description of the data rather than at the end. Be sure to mention the clinical setting for each trial:  e.g. distant metastatic disease, adjuvant therapy for resected regionally advanced disease, etc.  Do not imply that the systemic therapies are considered alternatives to surgery for localized disease.

Immunotherapy of cutaneous melanoma: Lines 155-163 and 163-171 are redundant—exact same wording. One set should be deleted.

Immunotherapy of vulvovaginal melanoma: Lines 233-240.  Authors should also comment on the levels of PDL1 expression, and define what is meant by positive PDL-1 expression in lines 238-239.

Conclusions: line 276. I would question whether treatment options for VV melanoma are limited compared to cutaneous.  It appears to me that depending on stage of disease, they are both treated with surgery, especially for local disease, radiation for certain regionally advanced and metastatic lesions, and systemic therapies: immune checkpoint inhibitor immunotherapy, BRAF/MEK inhibitors for patients with relevant BRAF mutations, and interleukin-2 and chemotherapy in certain situations.

Would suggest restating these key concepts:

·      VV melanomas are much less common that cutaneous, and therefore less well understood.

·      Major genomic differences include typical numbers of non-synonymous mutations, which likely makes VV less responsive to immune checkpoint immunotherapy, and increased frequencies of specific driver mutations, which may explain relatively more aggressive behavior in terms of more advanced stage at diagnosis and poorer survival within similar stages of disease, compared to cutaneous melanoma.

·      Modern immunotherapy is more efficacious in cutaneous than VV, presumably because of numbers of immunogenic targets that result from much greater numbers of non-synonymous mutations in the genomes of cutaneous melanomas.

Author Response

(The authors gave the same response as above.)

Round 2

Reviewer 1 Report

The manuscript is much improved with the revision. Thank you for accepting my suggestions and answering all my questions.

Minor Points:

1. (Pg 2, lane 49-52 – Introduction section)— “Cutaneous melanoma can be divided into 4 distinctive main types – superficial” – Types should be corrected as – “Superficial spreading, Nodular, Lentigo Maligna and Acral lentiginous”.

2. Typo - (Pg 3, lane 82)—“ such as BRAF, NRAS, PT53, CDKN2A and PTEN” – Please correct TP53.

3. Typo - (Pg 3, lane 83)—“ BRAF – _gene mutation (variant Val600)” – Please correct variant V600E.

4. (Pg 3, lane 83-86 – 2 sentences)—“ BRAF – _gene mutation (variant Val600) are found in roughly 50% of primary skin melanoma, however the role of this mutation in case of metastatic melanoma is yet to be determined. According to some studies the percentage of primary melanoma rises up to 60% and amounts to similar numbers in metastases of melanoma [18].” – The second sentence is probably redundant or please rephrase this sentence.

5. Typo - (Pg 4, lane 116)—“ P13 mTOR kinase” – Please correct PI3K-mTOR.

6. (Pg 9, lane 273-274)—“ What is interestingly is that KIT mutation rate was higher in vulvar melanoma (31.4) than in vulvar melanoma (6.2%).” – Vulvar is twice – I believe you meant “vaginal”.

7. Typo - (Pg 9, lane 310)—“ better over survival” – Please correct “overall”.

8. Typo - (Pg 10, lane 314)—“ metastis during the treatment” – Please correct “metastasis”.

9. Typo - (Pg 10, lane 315)—“ The response rate reached 28,5%.” – Please correct “28.5”. I understand , and . are used interchangeably in many countries. Please keep it “.” (such as 28.5” in the whole manuscript and tables for consistency and better understanding.

Author Response

Dear Reviewer,

We send our response to your comments.

Kind regards,

Anna Lorenz

corresponding author
